# Fluorescent Lymphography-Guided Lymphadenectomy during Minimally Invasive Completion Total Gastrectomy for Remnant Gastric Cancer Patients

**DOI:** 10.3390/cancers14205037

**Published:** 2022-10-14

**Authors:** Nasser Alrashidi, Ki-Yoon Kim, Sung Hyun Park, Sejin Lee, Minah Cho, Yoo Min Kim, Hyoung-Il Kim, Woo Jin Hyung

**Affiliations:** 1Department of Surgery, Yonsei University College of Medicine, Seoul 03722, Korea; 2Gastric Cancer Center, Yonsei Cancer Center, Yonsei University Health System, Seoul 03722, Korea; 3Department of Surgery, Unaizah College of Medicine and Medical Sciences, Qassim University, Buraydah P.O. Box 6688, Al-Qassim, Saudi Arabia

**Keywords:** minimally invasive completion total gastrectomy, remnant gastric cancer, lymphadenectomy, fluorescent lymphography

## Abstract

**Simple Summary:**

The altered lymphatic anatomy around the remnant stomach after initial surgery causes technical difficulties in systematic lymphadenectomy during the completion total gastrectomy. A fluorescent lymphography with indocyanine green under near-infrared imaging is a reliable intraoperative technique for lymphatic identification in minimally invasive gastric cancer surgery. This study aimed to assess the clinical application of fluorescent lymphography in minimally invasive completion total gastrectomy for remnant gastric cancer. More lymph node retrieval was demonstrated in minimally invasive completion total gastrectomy with fluorescent lymphography than without fluorescent lymphography. Fluorescent lymphography is an effective tool for the intraoperative assessment of lymphatics around the remnant stomach and systemic lymphadenectomy during minimally invasive completion total gastrectomy.

**Abstract:**

No study has evaluated fluorescent lymphography for lymphadenectomy in remnant gastric cancer (RGC). This study aimed to assess the clinical application of fluorescent lymphography in minimally invasive completion total gastrectomy for RGC. Patients who had undergone minimally invasive completion total gastrectomy for RGC from 2013 to 2020 were retrospectively reviewed. The perioperative outcomes and long-term prognosis were compared between patients who had undergone minimally invasive completion total gastrectomy with fluorescent lymphography (the FL group) and those without fluorescent lymphography (the non-FL group). The FL group comprised 32 patients, and the non-FL group comprised 36 patients. FL visualized lymphatics in all 32 patients without complications related to the fluorescent injection. The median number [the interquartile range] of LN retrieval was significantly higher in the FL group (17 [9.3–23.5]) than in the non-FL group (12.5 [4–17.8]); *p* = 0.016). The sensitivity of fluorescent lymphography in detecting metastatic LN stations was 75%, and the negative predictive value was 96.9% in the FL group. The overall relapse-free survivals were comparable between the groups (*p* = 0.833 and *p* = 0.524, respectively). FL is an effective tool to perform a more thorough lymphadenectomy during minimally invasive completion total gastrectomy for RGC. Using FL in RGC surgery may improve surgical quality and proper staging.

## 1. Introduction

Remnant gastric cancer (RGC) is a malignancy arising in the residual stomach following partial gastrectomy for benign or malignant gastric diseases [1,2,3]. An increase in the long-term survival of gastric cancer patients resulted from early detection through screening and regular postoperative endoscopic follow-ups after initial partial gastrectomy for primary gastrectomy influenced the rising incidence of RGC in recent years [4,5,6,7,8]. The treatment choice for RGC is completion total gastrectomy with systematic lymphadenectomy [9,10,11]. The lymphatic flow around the remnant stomach is altered because of changes in the anatomy after the initial surgery [12,13,14]. These alterations vary according to the resection extent, the inclusion of lymphadenectomy, the type of reconstruction of the initial surgery, and the degree of adhesion after surgery. Thus, lymphadenectomy during completion total gastrectomy should be differentiated based on the characteristics of the initial surgery and altered lymphatic anatomy around the remnant stomach. Consequently, along with the technical difficulties in systematic lymphadenectomy during completion total gastrectomy, surgeons have concerns about the oncological safety related to the completeness of lymphadenectomy during minimally invasive completion total gastrectomy [15,16]. Thus, a tool that allows visualization of the lymphatics around the remnant stomach would benefit the surgeon to perform adequate and complete lymphadenectomy during the completion total gastrectomy.

Fluorescent lymphography (FL) with indocyanine green (ICG) under near-infrared imaging has been introduced as a reliable intraoperative technology for lymphatic identification in minimally invasive gastric cancer surgery [17,18,19,20,21,22]. FL can lead to complete lymphadenectomy during minimally invasive surgery for gastric cancer. Therefore, in the present study, we hypothesized that FL would facilitate lymphadenectomy during minimally invasive completion total gastrectomy for RGC patients. To the best of our knowledge, no study has evaluated the potential role and significance of FL during minimally invasive completion total gastrectomy for RGC. This study aimed to assess the clinical application of FL in minimally invasive completion total gastrectomy for RGC by comparing the perioperative outcomes and long-term prognosis of patients who had undergone minimally invasive completion total gastrectomy with or without FL.

## 2. Materials and Methods

### 2.1. Patients

A retrospective review of a prospectively collected database on remnant gastric cancer patients who had undergone minimally invasive completion total gastrectomy with systemic lymphadenectomy between April 2013 and December 2020 was performed at a tertiary hospital. All the patients had undergone previous distal gastrectomy for gastric cancer or benign diseases. All the patients were diagnosed with adenocarcinoma in the remnant stomach by preoperative endoscopy with biopsy. Patients who had received preoperative chemotherapy or radiation treatment were excluded from the study. Our institution generally recommends minimally invasive surgery for patients without evidence of adjacent organ invasion (clinical T4b) or extensive lymph node metastasis to extra-perigastric lymph nodes on the preoperative assessment. The same strategy is applied when performing minimally invasive completion total gastrectomy for RGC. Thus, patients eligible for minimally invasive completion total gastrectomy were informed of the operative procedures, possible conversion to other approaches, costs, and risks associated with laparoscopic and robotic approaches. After being fully informed, all the patients chose which type of operation they would receive and provided written informed consent for surgery before the operation. The Institutional Review Board of Severance Hospital, Yonsei University College of Medicine, Seoul, Korea, approved the study and waived the informed consent for the use of patient data because of its retrospective nature (IRB number: 4-2022-0330).

Patients who had undergone minimally invasive completion total gastrectomy with fluorescent lymphography-guided lymphadenectomy (the FL group) were compared with those who had undergone minimally invasive completion total gastrectomy without fluorescent lymphography (the non-FL group). Data regarding the patients’ demographics and operative information, including the use of FL, pathology, and the survival and recurrence statuses, were collected from the electronic medical records system. Additionally, data regarding a previous gastrectomy were also collected, such as the indication for initial surgery (benign disease or cancer), surgical approach, and type of anastomosis (gastroduodenostomy or gastrojejunostomy).

### 2.2. Endoscopic Injection of ICG

The detailed endoscopic ICG injection protocol for FL-guided lymphadenectomy was reported previously [18,19,20]. Since FL-guided lymphadenectomy requires an ICG injection 1 day before surgery, a submucosa injection instead of a subserosal injection was performed. From 2013 to 2014, 0.6 mL of an ICG (Dongindang Pharmaceutical Co., Siheung, Korea) solution concentrated in 1.25 mg/mL was endoscopically injected into the submucosal layer of the stomach at four points around the primary lesion with a total volume of 2.4 mL (total of 3 mg of ICG) during endoscopy for tumor identification on the day before surgery. After the adoption of a da Vinci^®^ Xi system (Intuitive Surgical, Sunnyvale, CA, USA) and a Pinpoint^®^ fluorescence imaging system (Novadaq, Mississauga, Ontario, Canada) in 2015, the concentration of the ICG solution was diluted to 0.625 mg/mL, and a total of 1.5 mg of ICG was injected because of the high sensitivity of the fluorescence signal intensity with those imaging systems.

Whenever possible, FL was used during minimally invasive completion total gastrectomy. In the following circumstances, the surgery was performed without FL: (1) the patient refused to undergo FL; (2) the endoscopic injection of indocyanine green was not possible on the day before surgery, such as if the operation day was on a Monday or the day after a holiday; (3) the near-infrared imaging system was not available.

### 2.3. Surgical Procedures Including Fluorescent Lymphography-Guided Lymphadenectomy

Detailed descriptions of our institutional procedures for robotic and laparoscopic completion total gastrectomy have been reported previously [23,24,25]. Laparoscopic or robotic completion total gastrectomy varied from the usual standard minimally invasive total gastrectomy for primary gastric cancer, particularly for adhesiolysis and lymphadenectomy. Briefly, the first trocar is inserted using an open technique in the right lower area to explore the abdominal cavity and identify the adhesions by introducing the camera through this trocar. The other trocars are placed in adhesion-free areas, and the release of adhesions to the abdominal wall is performed when necessary to allow insertion of the remaining trocars, similar to that used for usual minimally invasive gastrectomy. This maneuver helps to avoid bowel injuries that may occur during trocar insertion related to adhesions caused by the previous operation. The specific dissection required for completion total gastrectomy is usually to identify the plane between the liver and stomach for liver retraction. During robotic surgery, adhesiolysis can be performed using laparoscopic instruments to allow proper trocar placement, but most adhesiolysis procedures are preferably performed robotically to maximize the benefit of the robotic system. The type of the previous anastomosis determines the starting point for dissection. If the previous reconstruction was gastroduodenostomy, a greater curvature dissection is the starting point, and the dissection extends up to the esophagogastric junction to remove lymph node stations 4sa and 2. After full mobilization of the gastroduodenostomy area, the duodenum is divided. The celiac axis area is dissected unless this area was not dissected previously. After completion of the esophageal mobilization, the lower esophagus is transected, and the specimen is collected in a plastic bag. If the previous reconstruction was gastrojejunostomy, adhesions between the remnant stomach or anastomosis area and transverse colon area are released. Next, both the afferent and efferent jejunal loops are divided. After the division of the jejunum, the stomach can be retracted to perform lymphadenectomy and remnant stomach resection. After retrieval of the specimen, Roux-en-Y esophagojejunostomy is performed using either circular or linear staplers. At 45–50 cm distal to the esophagojejunostomy site, jejunojejunostomy is created using linear staplers.

When performing FL-guided lymphadenectomy, the surgical field is examined by switching from white light to near-infrared mode to detect fluorescent lymph nodes. The dissected surgical field was confirmed using near-infrared imaging to ensure complete lymphadenectomy by verifying no residual fluorescence in the dissected area, and any remaining fluorescent lymph nodes were removed.

### 2.4. Lymph Node Retrieval and Examination

Detailed descriptions of our institutional method of lymph node retrieval and examination after FL-guided lymphadenectomy have been reported previously [18,19,20,26,27,28]. A surgeon who participated in the surgery dissected the resected specimen and classified lymph node stations according to the Japanese classification of gastric carcinoma (3rd English edition) in the operation room [2]. In cases of FL-guided lymphadenectomy, the surgeon checked for fluorescence in each lymph node and lymph node station under near-infrared imaging. The lymph nodes at each station were then categorized as fluorescent or non-fluorescent lymph nodes based on the presence of fluorescence. A fluorescent station was defined as a lymph node station that contained at least one fluorescent lymph node, whereas a non-fluorescent station contained no fluorescent lymph nodes. After pathologic evaluation, the presence of fluorescence in paraffin-embedded lymph node blocks was re-confirmed using a near-infrared system.

### 2.5. Statistical Analysis

A Student’s *t*-test or Mann–Whitney U test was performed on continuous variables and means with standard deviation (SD) or medians with interquartile (IQR) were provided depending on the analysis methods. Chi-squared or Fisher’s exact probability tests were performed on categorical variables, and numbers with a percentage were presented. We calculated true-positive (TP, the number of fluorescent metastatic stations or lymph nodes), false-positive (FP, the number of fluorescent non-metastatic stations or lymph nodes), false-negative (FN, the number of non-fluorescent metastatic stations or lymph nodes), and true-negative (TN, the number of non-fluorescent non-metastatic station or lymph nodes) to evaluate the sensitivity, specificity, positive predictive value (PPV), and negative predictive value (NPV). Sensitivity was defined as TP/(TP + FN), specificity as TN/(TN + FP), PPV as TP/(TP + FP), and NPV as TN/(TN + FN). The Kaplan–Meier method was used to estimate survival curves. A log-rank test was used to assess differences between survival curves. All tests were two-sided, and statistical significance was considered as a *p* value < 0.05. IBM SPSS Statistics software for Windows, version 26.0, was used to conduct the statistical analyses (IBM Corp., Armonk, New York, NY, USA).

## 3. Results

Subsection

Comparison of perioperative outcomes;Comparison of lymph node retrieval;Diagnostic accuracy of fluorescent lymphography-guided lymphadenectomy;Comparison of survival.

A total of 71 patients had undergone minimally invasive total completion gastrectomy for RGC during the study period. After excluding three patients who had received neoadjuvant chemotherapy, thirty-two patients comprised the FL group, and thirty-six patients comprised the non-FL group. ICG was successfully injected endoscopically on the day before surgery without technical failures and any adverse events in all 32 patients in the FL group. Fluorescent lymphography well-visualized all draining lymph nodes from the primary lesion in all 32 patients in the FL group (Figure 1). During the operation, no intraoperative adverse events associated with the use of near-infrared fluorescent imaging were observed.

### 3.1. Comparison of Perioperative Outcomes

No differences were found in the demographic features between the groups, such as age, sex, BMI, or the physical status classification of the American Society of Anesthesiologists. The two groups were similar in terms of clinical stage, tumor size, pathological T classification, and N classification (Table 1). However, a significant difference was found in the operation method (62.5% of robotic surgery in the FL group and 22.2% of robotic surgery in the non-FL group; *p* = 0.001). No statistical differences were found in the mean operation time and estimated blood loss between the groups. The rate of all complications did not differ between the groups (*p* = 0.392), as well as the rate of complications grade IIIa or higher (15.0% in the FL group and 15.4% in the non-FL group). Postoperative mortality was not found in either group. The median postoperative hospital stay did not differ between the groups (FL: 6 days [IQR, 5–10.5] vs. non-FL: 7.5 days [IQR, 6–15]; *p* = 0.099) (Table 2).

### 3.2. Comparison of Lymph Node Retrieval

We identified 146 dissected stations having 630 retrieved lymph nodes in 32 patients in the FL group. Of these, there were 114 (78.1%) fluorescent lymph node stations having 455 (72.2%) fluorescent lymph nodes. The median numbers of the retrieved lymph nodes and dissected stations were 17 (IQR, 9.3–23.5) and 4 (IQR, 3–6) per patient, respectively. The median numbers of fluorescent lymph nodes and stations were 10.5 (IQR, 6–16.5) and 3.5 (IQR, 2–4.8) per patient, respectively.

The median number of lymph nodes retrieved in the FL group (17 [IQR, 9.3–23.5]) was significantly larger than in the non-FL group (12.5 [IQR, 4–17.8], *p* = 0.016). The median number of lymph node stations in the FL group (4 [IQR, 3–6]) was larger than in the non-FL group (3 [IQR, 2–5]), although there was no statistical difference (*p* = 0.126). In the subgroup analysis according to the FL, there were no differences in the retrieved LNs between laparoscopic and robot gastrectomy regardless of the FL (FL group: laparoscopy 12.5 [IQR, 7–20.5] versus robot 18 [IQR, 14.5–28.5], *p*-value 0.086. Non-FL group: laparoscopy 13.5 [IQR, 4–18] versus robot 11.5 [IQR, 4.5–15.5), *p*-value 0.717). Among the 55 patients who had undergone prior gastrectomy for primary gastric cancer, the median number of the retrieved lymph nodes was marginally larger in the FL group (*n* = 24, 15 [IQR, 7.3–18]) than in the non-FL group (*n* = 31; 11 [IQR, 4–15]; *p* = 0.056). Similarly, among the 13 patients who had a previous gastrectomy for a peptic ulcer, the median number of retrieved lymph nodes was also marginally higher in the FL group (*n* = 8, 28.5 [IQR, 21–52.3]) than in the non-FL group (*n* = 5, 19 [IQR, 17.5–24], *p* = 0.067). The proportion of patients with 16 or more lymph nodes retrieved was higher in the FL group (59.4%) than in the non-FL group (33.3%; *p* = 0.031). Patients with fewer than 16 lymph nodes retrieved and with lymph node metastasis were not identified in the FL group and were only identified in the non-FL group (16.7%; *p* = 0.026) (Table 3). The mean number of the retrieved lymph nodes by each station is shown in Figure 2.

### 3.3. Diagnostic Accuracy of Fluorescent Lymphography-Guided Lymphadenectomy

Four patients (12.5%) in the fluorescent group had pathologically positive LNs. Among these four patients with lymph node metastases, nine metastatic LNs in four LN stations were identified. Among these, three patients had one metastatic LN station in the fluorescent station at the left paracardial, lesser curvature, and splenic hilar LN station, respectively. The other patient had five metastatic but non-fluorescent LNs all in the non-fluorescent station at the right paracardial LN station. The patient with metastatic LNs in the non-fluorescent station had a 65-mm tumor size with an extensive lymphovascular invasion. In the non-fluorescent group, eight of thirty-six patients (22.2%) had LN metastases. Of these eight patients, five had perigastric LN metastases and three had LN metastases in both the perigastric and extra-perigastric stations.

In the FL-guided lymphadenectomy, the sensitivity for identifying LN metastasis based on FL was 44.4% (4 TP/9 TP + FN), and the specificity was 27.4% (170 TN/621 TN + FP). The PPV and NPV were 0.9% (4 TP/455 TP + FP) and 97.1% (170 TN/175 TN + FN), respectively. Based on the fluorescent stations, the sensitivity for detecting metastasis was 75% (1 TP/4 TP + FN), and the specificity was 21.8% (31 TN/142 TN + FP). The PPV was 2.6% (3 TP/114 TP + FP), and the NPV was 96.9% (31 TN/32 TN + FN) (Table 4).

### 3.4. Comparison of Survival

With the median follow-up of 35 months in the FL group and 37.5 months in the non-FL group, there was no significant difference in both the overall survival (*p* = 0.833) and the relapse-free survival (*p* = 0.524) between the FL and non-FL groups (Figure 3). During the follow-up, four patients died in the FL group, three of them had a cancer recurrence, and one had other diseases, while four patients died in the non-FL group, two of them had a cancer recurrence, and two had other diseases. There were four (12.5%) recurrences in the FL group, which were distant metastases, and there were seven (19.4%) recurrences in the non-FL group, which were one locoregional recurrence at the esophagojejunostomy anastomosis site and six distant metastases.

## 4. Discussion

FL by the endoscopic peritumoral submucosal injection of ICG administered 1 day before surgery successfully visualized the lymph nodes draining from the primary tumor in the remnant stomach. FL-guided lymphadenectomy during the minimally invasive completion total gastrectomy for RGC demonstrated a significantly larger number of LNs than without FL-guided lymphadenectomy. Additionally, the negative predictive value was 96.9% if the LN stations were non-fluorescent under FL, although the sensitivity was 75% for detecting the metastatic LN stations. FL allowed a more thorough lymphadenectomy and examined the LN status more meticulously, although the survival was not affected.

Systemic lymphadenectomy for gastric cancer is a technically demanding procedure. Additionally, lymphadenectomy during completion total gastrectomy for RGC is even more difficult because of the rarity of remnant cancer and the distorted lymphatic anatomy around the remnant stomach. Cancer in the remnant stomach may spread to the hepatoduodenal ligament, superior mesenteric vein, and splenic and short gastric vessels after gastroduodenostomy, while it may spread to the splenic and short gastric vessels and jejunal mesentery through the anastomosis site after gastrojejunostomy [12,13,14]. Cancer in the remnant stomach can spread through the lymphatics around the remaining vessels if the right gastroepiploic, right gastric, and left gastric vessels are retained after previous surgery. By visualizing the lymphatics and lymph nodes around the remnant stomach using FL, surgeons may perform a safer, easier, and more effective lymphadenectomy, even with the altered lymphatic anatomy caused by the prior gastrectomy.

Consistent with previous studies, FL-guided lymphadenectomy during minimally invasive completion total gastrectomy resulted in a larger number of retrieved LNs around the remnant stomach without any adverse effect in the current study [17,18,19,20]. The adequate number of retrieved LNs is essential for proper nodal staging. More lymph node examinations allow for a more accurate determination of lymph node metastasis. Regarding RGC, no standardized nodal staging criteria are available because the number of lymph nodes retrieved did not exceed 16 in most cases [15,16]. In the present study, FL-guided lymphadenectomy showed 16 or more lymph nodes retrieved in 59.6% of patients, suggesting that the nodal staging system for primary gastric cancer can be applied to almost 60% of RGC patients.

When performing lymphadenectomy during completion total gastrectomy, challenges primarily occur in the extra-perigastric areas, particularly in the splenic hilar area. Splenic hilar area dissection is a recommendation based on a relatively high therapeutic index and splenic hilar LN metastasis rates of 14.1% and 19.2% in the RGC after partial gastrectomy for malignant and benign disease, respectively [29,30,31]. FL enabled the intraoperative assessment of the quality of the lymphadenectomy because any fluorescent lymph nodes remaining in the dissected area would be clearly identified, and the risk of leaving LNs in the dissected area would be decreased. Additionally, FL has been suggested as a useful tool to perform high-quality lymphadenectomy at the splenic hilum in total gastrectomy for primary gastric cancer when fluorescence is present. Thus, FL for RGC is a promising technique for thorough splenic hilar area dissection during completion total gastrectomy. Furthermore, the LN metastasis rate in the splenic hilar area in advanced RGC has been reported to be less than 20%; over 80% of RGC patients would receive an unnecessary splenic hilar area dissection. Considering the high negative predictive value (over 95%) of FL, FL might be employed with caution as a tool to omit splenic hilar dissection during completion total gastrectomy when fluorescence is absent, particularly for patients with high surgical risk. However, one patient had a false-negative lymph node station. This patient had a large tumor size and extensive lymphovascular invasion. Although FL shows the lymphatic flow from the primary tumor, visualization may fail if a lymphatic flow obstruction occurs because of large tumor size or extensive lymphovascular invasion.

The present study has limitations. The small number of patients analyzed because of the rarity of RGC can limit more in-depth statistical analyses of the specificity and sensitivity of LN metastasis detection with FL. Because minimally invasive completion total gastrectomy was applied mostly for earlier-stage cancer, only 12.5% of patients in the fluorescent group had metastatic lymph nodes. This rate is relatively low compared with that in previous studies for RGC. Thus, a low rate of LN metastasis in this study prohibits the risk factor analysis of false-negative results. Furthermore, the risk factors for FL false-negative results are unknown; future studies of these risk factors are warranted. Another limitation is related to ICG. Because ICG is a non-tumor-specific tracer, the specificity and positive predictive value of the FL were low. Thus, selective lymphadenectomy based on FL with ICG instead of systemic lymphadenectomy warrants further study.

## 5. Conclusions

FL is an effective tool to perform a more thorough lymphadenectomy during minimally invasive completion total gastrectomy with the help of the visualization of the draining lymphatics and lymph nodes from the primary tumor in the remnant stomach. FL may help decide the effective lymphadenectomy extent during CTG. Based on the findings of the current study, using FL with near-infrared imaging in RGC surgery may improve surgery quality and proper staging.

## Figures and Tables

**Figure 1 cancers-14-05037-f001:**
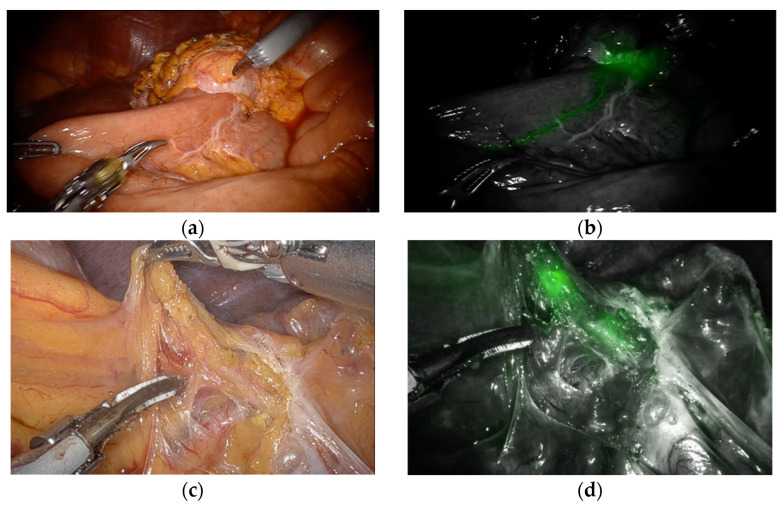
(**a**) White-light image of RGC after distal gastrectomy with gastrojejunostomy; (**b**) fluorescent imaging of “Figure 1a” showing lymphatic drainage to jejunal mesentery from the tumor located in the anastomosis site; (**c**) white-light image of distal splenic artery area in remnant gastric cancer; (**d**) fluorescent image of “Figure 1c” showing fluorescent LNs along the distal splenic artery.

**Figure 2 cancers-14-05037-f002:**
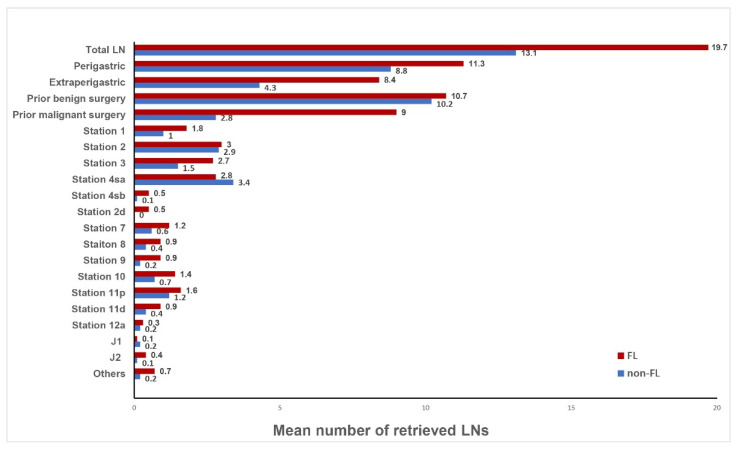
Mean number of retrieved lymph nodes.

**Figure 3 cancers-14-05037-f003:**
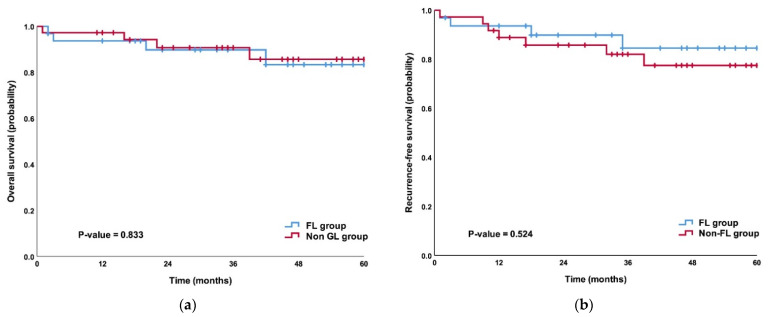
Kaplan–Meier estimates of the probability of all patients’ (**a**) overall survival and (**b**) relapse-free survival.

**Table 1 cancers-14-05037-t001:** Comparison of clinicopathologic characteristics.

	FL Group (*n* = 32)	Non-FL Group (*n* = 36)	*p*-Value
Age (years), mean (SD)	62.5 (14.7)	59.9 (14.1)	0.452
Sex			0.265
Male	21 (65.6)	28 (77.8)	
Female	11 (34.4)	8 (22.2)	
BMI (kg/m^2^), median (IQR)	21.9 (6.3)	21.4 (3.9)	0.636
ASA classification			0.622
I	2 (6.3)	6 (16.7)	
II	17 (53.1)	17 (47.2)	
III	11 (34.4)	11 (30.6)	
IV	2 (6.3)	2 (5.6)	
Prior gastrectomy			0.890
STG with BI	10 (31.3)	11 (30.6)	
STG with BII	21 (65.6)	25 (69.4)	
STG with RY GJ	1 (3.1)	0	
Operation method of prior gastrectomy			0.226
Open	13 (40.6)	18 (50.0)	
Laparoscopy	9 (28.1)	13 (36.1)	
Robot	10 (31.3)	5 (13.9)	
Cause of prior gastrectomy			0.245
Cancer	24 (75.0)	31 (86.1)	
Peptic ulcer	8 (25.0)	5 (13.9)	
Clinical T classification †			0.829
T1	25 (78.1)	31 (86.1)	
T2	4 (12.5)	3 (8.3)	
T3	2 (6.3)	1 (2.8)	
T4	1 (3.1)	1 (2.8)	
Clinical N classification †			0.660
N0	29 (90.6)	34 (94.4)	
N+	3 (9.4)	2 (5.6)	
Tumor size (mm), mean (SD)	35.7 (19.5)	31.2 (18.9)	0.338
Pathologic T stage †			0.754
T1	21(65.6)	20(55.6)	
T2	4(12.5)	5(13.9)	
T3	2(6.3)	5 (13.9)	
T4a	5(15.6)	6(16.7)	
Pathologic N stage †			0.541
N0	28 (87.5)	28 (77.8)	
N1	3 (9.4)	3 (8.3)	
N2	1 (3.1)	2 (5.6)	
N3	0	3 (8.3)	

Values in parentheses are percentages. Abbreviations: FL, fluorescent lymphography; BMI, body mass index; ASA, American Society of Anesthesiologists; SD, standard deviation; IQR, interquartile range; STG with BI, subtotal gastrectomy with gastroduodenostomy; STG with BII, subtotal gastrectomy with gastrojejunostomy; STG with RY GJ, subtotal gastrectomy with Roux-en-Y gastrojejunostomy. † TNM stage according to AJCC 8th.

**Table 2 cancers-14-05037-t002:** Comparison of operative and postoperative outcomes.

	FL Group (*n* = 32)	Non-FL Group (*n* = 36)	*p*-Value
Operation method			0.001
Laparoscopy	12 (37.5)	28 (77.8)	
Robot	20 (62.5)	8 (22.2)	
Combined resection			0.660
No	29(90.6)	34 (94.4)	
Yes	3 (9.4)	2 (5.6)	
Operation time (minutes), mean (SD)	273.8 (77.0)	252.0 (58.2)	0.190
Estimated blood loss (mL), median (IQR)	100 (52–154)	115 (85–211)	0.224
Postoperative complications			0.392
Absent	12 (37.5)	10 (27.8)	
Present	20 (62.5)	26 (72.2)	
Clavien–Dindo Classification			0.466
Grade I	8 (40.0)	6 (23.1)	
Grade II	9 (45.0)	16 (61.5)	
≥Grade IIIa	3 (15.0)	4 (15.4)	
Postoperative mortality	0	0	0
Hospital stays (days), median (IQR)	6 (5–10.5)	7.5 (6–15)	0.099

Values in parentheses are percentages. Abbreviations: FL, fluorescent lymphography; SD, standard deviation; IQR, interquartile range.

**Table 3 cancers-14-05037-t003:** Comparison of retrieved lymph nodes.

Patients	FL Group (*n* = 32)	Non-FL Group (*n* = 36)	*p*-Value
Number of retrieved LN, median (IQR)	17 (9.3–23.5)	12.5 (4–17.8)	0.016
Number of retrieved LN in the patients who underwent prior gastrectomy due to cancer, median (IQR)	15 (7.3–18) (*n* = 24)	11 (4–15) (*n* = 31)	0.056
Number of retrieved LN in the patients who underwent prior gastrectomy due to peptic ulcer, median (IQR)	28.5 (21–52.3) (*n* = 8)	19 (17.5–24) (*n* = 5)	0.067
Number of cases that show total retrieved lymph nodes ≥ 16 (percentage)	19 (59.4%)	12 (33.3%)	0.031
Total retrieved lymph nodes <16 and metastatic lymph nodes ≥1 (percentage)	0	6 (16.7%)	0.026

Values in parentheses are percentages. Abbreviations: FL, fluorescent lymphography; LN, lymph node; IQR, interquartile range.

**Table 4 cancers-14-05037-t004:** Diagnostic value of fluorescent lymphography based on stations and lymph nodes.

	Total Number	Number of Metastasis	Number of Non-Metastasis	Sensitivity (%)	Specificity (%)	Positive Predictive Value (%)	Negative Predictive Value (%)
Total LN stations	146						
Fluorescent station	114	3	111	75	21.8	2.6	96.9
Non-fluorescent station	32	1	31				
Total LNs	630						
Fluorescent LNs	455	4	451	44.4	27.4	0.9	97.1
Non-fluorescent LNs	175	5	170				

FL, fluorescent lymphography; LN, lymph node.

## Data Availability

The data presented in this study are available upon request from the corresponding author.

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
