# Peer review of "Fluorescent Lymphography-Guided Lymphadenectomy during Minimally Invasive Completion Total Gastrectomy for Remnant Gastric Cancer Patients"

_cancers, 2022, doi:10.3390/cancers14205037_

Round 1

Reviewer 1 Report

This is a "one of its kind" paper which certainly deserves publication, being written in a fluent scientific English and corroborated by a sound statystical support. The numerosity of the data-set is outstanding and to my knowledge there are no so huge series worldwide of completion gastrectomies for remnant stomach cancer, performed robotically with fluorescent lympography. 

If I may suggest a slight change for the sake of adding technical interest to the manuscript, the Authors are asked to reconsider a shortening of their complete discussion and subsequent figures; at the same time a colour picture of the surgical field including lymphatic mapping around the gastric stump would be extremely appreciable.

Authors are anyway to be congratulated for such a stimulating unique report.

Author Response

Title: Fluorescent lymphography-guided lymphadenectomy during minimally invasive completion total gastrectomy for remnant gastric cancer patients

We appreciate your comprehensive and critical review of our manuscript, entitled "Fluorescent lymphography-guided lymphadenectomy during minimally invasive completion total gastrectomy for remnant gastric cancer patients" which we submitted as an Original Article for publication in the Cancers. We greatly appreciate the reviewers' helpful comments on our study, and we have revised our manuscript accordingly to address their concerns. We have incorporated all of the reviewers' comments in the revised manuscript and highlighted all revisions therein. The following outlines our responses to the reviewers' comments:

Responses to Reviewer 1 Comments

Point 1: If I may suggest a slight change for the sake of adding technical interest to the manuscript, the Authors are asked to reconsider a shortening of their complete discussion and subsequent figures; at the same time a colour picture of the surgical field including lymphatic mapping around the gastric stump would be extremely appreciable.

Response 1: We appreciate your comprehensive review of our manuscript. We agree with your opinion regarding a shortening of our discussion and a color picture of the surgical field, including lymphatic mapping. We revised the discussion contents on page 10, line 335, and page 11, line 370 to make the article more easily readable. We also added the white-lite and fluorescent imaging during the completion total gastrectomy after distal gastrectomy with gastrojejunostomy on page 6, line 268, to explain the visualization of fluorescent lymphography using ICG during operation. For your convenience, we highlighted the above sentences with red text in the revised manuscript.

Reviewer 2 Report

In the manuscript entitled” Fluorescent lymphography-guided lymphadenectomy during minimally invasive completion total gastrectomy for remnant gastric cancer patients” , authors conducted a retrospective review of a prospectively collected database on remnant gastric cancer patients who had undergone minimally invasive completion total gastrectomy with systemic lymphadenectomy patients. A total of 71 patients had undergone minimally invasive total completion gastrectomy for RGC during the study period. 32 patients comprised the FL group, and 36 patients comprised the non-FL group. Based on the findings of the current study, using FL with near-infrared imaging in RGC surgery may improve surgery quality and proper staging.

Authors performed a lot of work in this study. However, some concerns are needed to be further explained.

1, The best way of administration of ICG is the key factors affecting the effect of intraoperative NIR fluorescence imaging. In general, ICG administration includes subserous injection and submucosal administration. Please explain in detail the reasons for choosing peritual submucosal administration the day before surgery in this study.

2, In this study, fluorescent lymphography well visualized all draining lymph nodes from the primary lesion in all 32 patients in the FL group. It is recommended to show some fluorescent images of the surgery for the readers to learn.

3. The proportion of robotic surgeries was 62.5% in FL group, much higher than the 22.2% in the non-FL group. While there are literature reports that robotic surgery can increase the number of lymph nodes isolated, please analyze the impact of this factor on the results of this study.

4. Only 12.5% of patients in the fluorescent group had metastatic lymph nodes. So, please discuss the clinical value of ICG for these patients.

Author Response

Responses to Reviewer 2 comments

Point 1: The best way of administration of ICG is the key factors affecting the effect of intraoperative NIR fluorescence imaging. In general, ICG administration includes subserous injection and submucosal administration. Please explain in detail the reasons for choosing peritumoral submucosal administration the day before surgery in this study.

Response 1: We understand your concern regarding the ICG injection method. However, there are many results that submucosal injection is better than subserosa injection to visualize the lymphatics around the stomach.[1-5] Moreover, to perform fluorescent lymphography-guided lymphadenectomy, it is essential to inject ICG one day before surgery, thus subserosal injection is not possible in this setting. We added additional explanation in the Methods section on page 3, line 104: “Since FL guided lymphadenectomy requires ICG injection 1 day before surgery, ICG was injected into the submucosa instead of subserosa.” For your convenience, we highlighted the above sentences with red text in the revised manuscript.

References

  1. Kitagawa, Y.; Fujii, H.; Kumai, K.; Kubota, T.; Otani, Y.; Saikawa, Y.; Yoshida, M.; Kubo, A.; Kitajima, M. Recent advances in sentinel node navigation for gastric cancer: A paradigm shift of surgical management. J Surg Oncol 2005, 90, 147-151; discussion 151-142.
  2. Tajima, Y.; Yamazaki, K.; Masuda, Y.; Kato, M.; Yasuda, D.; Aoki, T.; Kato, T.; Murakami, M.; Miwa, M.; Kusano, M. Sentinel node mapping guided by indocyanine green fluorescence imaging in gastric cancer. Ann Surg 2009, 249, 58-62.
  3. Kong, S.-H.; Bae, S.-W.; Suh, Y.-S.; Lee, H.-J.; Yang, H.-K. Near-infrared fluorescence lymph node navigation using indocyanine green for gastric cancer surgery. The Journal of Minimally Invasive Surgery 2018, 21, 95-105.
  4. Cianchi, F.; Indennitate, G.; Paoli, B.; Ortolani, M.; Lami, G.; Manetti, N.; Tarantino, O.; Messeri, S.; Foppa, C.; Badii, B. et al. The clinical value of fluorescent lymphography with indocyanine green during robotic surgery for gastric cancer: A matched cohort study. J Gastrointest Surg 2020, 24, 2197-2203.
  5. Liu, M.; Xing, J.; Xu, K.; Yuan, P.; Cui, M.; Zhang, C.; Yang, H.; Yao, Z.; Zhang, N.; Tan, F. et al. Application of near-infrared fluorescence imaging with indocyanine green in totally laparoscopic distal gastrectomy. J Gastric Cancer 2020, 20, 290-299.

Point 2: In this study, fluorescent lymphography well visualized all draining lymph nodes from the primary lesion in all 32 patients in the FL group. It is recommended to show some fluorescent images of the surgery for the readers to learn.

Response 2: We agree with your recommendation that adding fluorescent images of the surgery would help readers better understand. We added the white-light and fluorescent images of a patient with remnant gastric cancer after distal gastrectomy with gastrojejunostomy on page 6, line 268, to explain the visualization of fluorescent lymphography using ICG during operation.

Point 3: The proportion of robotic surgeries was 62.5% in FL group, much higher than the 22.2% in the non-FL group. While there are literature reports that robotic surgery can increase the number of lymph nodes isolated, please analyze the impact of this factor on the results of this study.

Response 3:  We appreciate your concern about the LN harvesting effect of robotic surgery. In the beginning, fluorescent-guided lymphadenectomy was only possible in robotic surgery, so the proportion of robotic surgery was higher in the FL group. However, in the subgroup analysis based on the fluorescent image use, there were no statistical differences in the retrieved LNs between laparoscopic and robotic surgery. We added these results from line 222 on page 5: “In the subgroup analysis according to FL, there were no differences in the retrieved LN between laparoscopic and robot gastrectomy regardless of FL.(FL group; laparoscopy 12.5 [IQR, 7-20.5] versus robot 18 [IQR, 14.5-28.5], p-value 0.086. Non-FL group; laparoscopy 13.5 [IQR, 4-18] versus robot 11.5 [IQR, 4.5-15.5), p-value 0.717)” For your convenience, we highlighted the above sentences with red text in the revised manuscript.

Point 4: Only 12.5% of patients in the fluorescent group had metastatic lymph nodes. So, please discuss the clinical value of ICG for these patients.

Response 4: As you mentioned, the lymph node metastasis rate in the fluorescent group was relatively low. Since minimally invasive completion total gastrectomy was applied to early-stage remnant gastric cancer, the lymph node metastasis rate is relatively low in our study. We already described this as a limitation in the Discussion section on page 10 as follows “The small number of patients analyzed because of the rarity of RGC can limit more in-depth statistical analyses of the specificity and sensitivity of LN metastasis detection with FL. Because minimally invasive completion total gastrectomy was applied mostly for earlier-stage cancer, only 12.5% of patients in the fluorescent group had metastatic lymph nodes. This rate is relatively low compared with that in previous studies for RGC.”